# Improved Adipose Tissue Function after Single Anastomosis Duodeno-Ileal Bypass with Sleeve-Gastrectomy (SADI-S) in Diet-Induced Obesity

**DOI:** 10.3390/ijms231911641

**Published:** 2022-10-01

**Authors:** Sara Becerril, Carlota Tuero, Javier A. Cienfuegos, Amaia Rodríguez, Victoria Catalán, Beatriz Ramírez, Víctor Valentí, Rafael Moncada, Xabier Unamuno, Javier Gómez-Ambrosi, Gema Frühbeck

**Affiliations:** 1Metabolic Research Laboratory, Clínica Universidad de Navarra, 31008 Pamplona, Spain; 2CIBER Fisiopatología de la Obesidad y Nutrición (CIBEROBN), Instituto de Salud Carlos III, 28029 Madrid, Spain; 3Obesity and Adipobiology Group, Instituto de Investigación Sanitaria de Navarra (IdiSNA), 31008 Pamplona, Spain; 4Department of Surgery, Clínica Universidad de Navarra, 31008 Pamplona, Spain; 5Department of Anesthesia, Clínica Universidad de Navarra, 31008 Pamplona, Spain; 6Medical Engineering Laboratory, University of Navarra, 31008 Pamplona, Spain; 7Department of Endocrinology & Nutrition, Clínica Universidad de Navarra, 31008 Pamplona, Spain

**Keywords:** sleeve gastrectomy, single anastomosis duodenoileal bypass, diet-induced obesity, brown adipose tissue, beige adipose tissue

## Abstract

Bariatric surgery has been recognized as the safest and most effective procedure for controlling type 2 diabetes (T2D) and obesity in carefully selected patients. The aim of the present study was to compare the effects of Sleeve Gastrectomy (SG) and Single Anastomosis Duodenoileal Bypass with SG (SADI-S) on the metabolic profile of diet-induced obese rats. A total of 35 four-week-old male Wistar rats were submitted to surgical interventions (sham operation, SG and SADI-S) after 4 months of being fed a high-fat diet. Body weight, metabolic profile and the expression of molecules involved in the control of subcutaneous white (SCWAT), brown (BAT) and beige (BeAT) adipose tissue function were analyzed. SADI-S surgery was associated with significantly decreased amounts of total fat pads (*p* < 0.001) as well as better control of lipid and glucose metabolism compared to the SG counterparts. An improved expression of molecules involved in fat browning in SCWAT and in the control of BAT and BeAT differentiation and function was observed following SADI-S. Together, our findings provide evidence that the enhanced metabolic improvement and their continued durability after SADI-S compared to SG rely, at least in part, on the improvement of the BeAT phenotype and function.

## 1. Introduction

Obesity is considered a chronic relapsing disease process associated with several pathophysiological changes including the progressive increase in insulin resistance along with the defect in insulin secretion, constituting a major risk factor for the development of type 2 diabetes (T2D) and cardiovascular diseases [1]. Bariatric surgery is widely considered to be the most effective treatment for severe obesity and its multiple comorbidities for carefully selected patients, being more efficient than conventional therapies, such as calorie restriction, lifestyle modification and pharmacotherapy [2,3,4].

Surgical procedures for weight loss include mainly three types of procedures that affect restriction, malabsorption or both. Gastric plication and sleeve gastrectomy (SG) reduce the gastric volume, resulting in caloric restriction. The jejuno-ileal bypass removes parts of the small intestine decreasing the absorption of ingested nutrients. Mixed interventions combine both restrictive and malabsorptive techniques. The Roux-en-Y gastric Bypass (RYGB), biliopancreatic diversion with duodenal switch and single anastomosis duodenoileal bypass with SG (SADI-S) combine both restrictive and malabsorptive techniques. Interventions affecting restriction trigger delayed glycemic control mainly related to weight loss, while malabsorptive procedures result in an early T2D remission as well as energy homeostasis reestablishment after surgery prior to significant weight loss has occurred [5].

Weight loss and glycemic effects derived from bariatric surgery were traditionally thought to be the result of caloric restriction (reduced gastric volume) and/or malabsorption of ingested nutrients, but these modifications do not fully account for the magnitude of weight loss observed after surgery. Recent studies have demonstrated that changes in energy balance physiology and adipose tissue (AT) mass are the primary mechanisms [6,7], suggesting that molecular mechanisms affecting metabolism underlie the effect of bariatric surgery. AT can be classified into white, brown or beige fat. While white adipose tissue (WAT) is a heterogeneous tissue mainly comprised of lipid-filled adipocytes that store energy reserves as fat, brown (BAT) and beige ATs (BeAT) are highly active metabolic organs specialized in non-shivering thermogenesis [8].

Previous studies of our group have shown that sleeve-gastrectomized rats exhibit improved excess adiposity as well as metabolic profile in experimental models of genetic (leptin-receptor deficient Zucker *fa*/*fa* rats) and diet-induced obesity (DIO) [9,10]. These beneficial effects of SG on energy homeostasis are mediated, at least in part, by the improvement of BAT function through the upregulation of UCP-1 in brown adipocytes [11]. We hypothesized that given the additional anatomo-physiological changes, the SADI-S procedure exerts a superior metabolic improvement when compared to the SG. In this context, our main objective was to compare the metabolic outcomes of SG and SADI-S as well as their impact on WAT, BAT and BeAT function in an animal model of DIO.

## 2. Results

### 2.1. Effect of SG and SADI-S on Adiposity, Energy Expenditure and Metabolic Profile

As expected, high-fat diet (HFD) induced an increase (*p <* 0.001) in body weight and whole-body white adiposity and impaired metabolic profile (Appendix A). Weight loss in rats submitted to SG and SADI-S was rapid, early and significant during the first week after the surgical interventions (Figure 1a). After this initial period, weight loss started to diverge becoming statistically significant in the SADI-S group. Sham-operated and sleeve gastrectomized rats gained weight, with sham animals returning to their preoperative body weight 6 weeks after surgery. The SADI-S group showed a significantly (*p <* 0.001) lower body weight from day 10 after surgery onwards, experimenting the maximal weight loss compared to their age-matched sham- and SG-operated counterparts at the end of the study. The final body weight of SADI-S rats was significantly (*p <* 0.001) lower than that of sham group (Table 1). The reduction in body weight in SADI-S rats was associated with significant reductions in epididymal (*p <* 0.001), subcutaneous (*p <* 0.001), perirenal (*p <* 0.001) and total fat depots (*p <* 0.001), as compared to SG group (Table 1). Neither liver nor skeletal muscle weights were statistically different among groups.

Basal rectal temperature was analyzed revealing no differences in energy expenditure of SADI-S rats compared with sham-operated animals (Table 1). Sleeve-gastrectomized rats exhibited significantly (*p* < 0.05) increased values compared to the sham-operated group, according to previous studies [11]. The increased activation of interscapular BAT of SG rats was further confirmed by infrared thermography (Table 1 and Figure 1b).

Oxygen consumption (VO_2_) and carbon dioxide production (VCO_2_) were measured in metabolic cages revealing that SG rats exhibited a significantly (*p* < 0.001) larger CO_2_ production than the SADI group with no differences in VO_2_, which is likely due to the oxidation of carbohydrates, resulting in greater production of CO_2_ in sleeve-gastrectomized rats (Figure 1c, d). These data were corroborated by the respiratory quotient (RQ), which was decreased in rats submitted to SADI-S, with values near 0.7 (0.69), indicating a shift from glucose to free fatty acids (FFA) as energy substrate and suggesting a mobilization of fat stores. A slightly increased RQ in SG rats (*p* = 0.051) compared to the sham group was also observed, with values near 1.0, confirming the carbohydrate oxidation [12]. This preferential lipid utilization by SADI-S rats was supported by the calculation of glucose (GOX) and lipid (LOX) oxidation, which corroborates that animals mobilize fat stores, providing fatty acids for oxidation and glycerol for gluconeogenesis (Figure 1e–g).

The metabolic profile of the experimental groups is shown in Table 1. Rats submitted to SADI-S exhibited an increased insulin sensitivity, as evidenced by a lower glycemia (*p* < 0.001), insulinemia (*p* < 0.01) and HOMA index (*p* < 0.001) as well as a higher (*p* < 0.001) QUICKI index compared to SG rats. SADI-S, but not SG, also improved the lipid profile, evidenced by reduced plasma levels of triglycerides (TG) and total cholesterol (both *p* < 0.01). In parallel to the decreased adiposity, the plasma levels of the adipokines leptin and adiponectin were significantly lower (*p* < 0.001 for each) in SADI-S rats as compared with the sham and SG groups. Importantly, the Adpn/Lep ratio, suggested as a biomarker of dysfunctional AT [13], significantly increased (*p* < 0.001) after weight loss in rats submitted to the SADI-S (Table 1).

### 2.2. Impact of the Bariatric Surgery on Markers of Brown Adipocyte Function

To gain further insight into the mechanisms underlying the improved energy expenditure of rats submitted to bariatric surgery, the gene expression levels of key molecules involved in BAT differentiation (Bmp7), function (Prdm16, Pgc1a, Pparg, Sirt1, Sirt3 and Sirt6) and regulation (Xbp1, Fgf21) were analyzed. Transcript levels of Bmp7, a factor involved in the activation of the brown adipogenesis program, were significantly increased (*p* < 0.05) after SADI-S, compared to sham-operated rats, and exhibited a tendency towards an increase vs. SG animals (*p* < 0.1). No changes in the mRNA expression levels of Prdm16, a zinc-finger protein that stimulates brown fat-selective gene expression, were observed, while the SADI-S surgery significantly increased (*p* < 0.05) Pgc1a gene expression levels as compared with the SG procedure. Moreover, transcript levels of Pparg, a mediator of brown adipocyte functionality, were significantly increased (*p* < 0.05) in both SG and SADI-S rats (Figure 2a–d). The gene expression levels of the sirtuin family members Sirt1, Sirt3 and Sirt6 in BAT was also determined, in order to delve whether the expression of these NAD+-dependent deacetylases is altered by bariatric surgery. As shown in Figure 2e–g, the mRNA levels of Sirt1, Sirt3 and Sirt6 exhibit a similar pattern of increase in BAT of SADI-S rats. No increase in the mRNA expression levels of the thermogenic genes Ucp1, Ucp3 or Dio2 in SADI-S rats was observed (Figure 2l–n).

The essential role of XBP1 in the transcriptional induction of Ucp1 in brown adipocytes has been recently described [14]. SADI-S rats exhibited significantly (*p* < 0.05) increased *Xbp1* gene expression levels as compared to SG counterparts (differences vs. sham-operated rats did not reach statistical significance, *p* = 0.071) (Figure 2h). The mRNA expression levels of the key regulator of the differentiation to brown adipocytes *Fgf21* was also upregulated (*p* < 0.05) in rats submitted to SADI-S as compared to the sham surgery (Figure 2i). Furthermore, the transcript levels of the apoptosis signal-regulating kinase 1 (*Ask1*), a key regulator of brown adipocyte gene expression and function [15], was also significantly increased in SADI-S animals as compared to the sham and SG groups (*p* < 0.05 and *p* < 0.001, respectively) (Figure 2j).

After SADI-S, a positive correlation between mRNA levels of *Sirt1* (r = 0.66; *p* < 0.01), *Pgc1a* (r = 0.43; *p* < 0.10), *Pparg* (r = 0.47; *p* < 0.05), *Bmp7* (r = 0.60; *p* < 0.01) and *Ask1* (r = 0.64; *p* < 0.01) with the relative rectal temperature was found (Appendix A).

### 2.3. Effect of SG and SADI-S on Factors Involved in Fat Browning in Subcutaneous Adipose Tissue

SCWAT adipocyte size from the HFD-fed mice was greater than that of the normal chow-fed littermates (5887 ± 356 µm^2^ vs. 4272 ± 493 µm^2^; *p* < 0.01). Six weeks after SADI-S surgery, undersized adipocytes as compared to sham-operated and sleeve-gastrectomized rats were detected (*p* < 0.001 and *p* < 0.05, respectively) (Figure 3a). Moreover, immunohistological analyses showed a higher expression of UCP-1 in SCWAT after SADI-S surgery (Figure 3b). In line with this observation, a tendency towards a decrease in *Slc27a1* (solute carrier family 27 member 1) gene expression levels was observed in SCWAT of SADI-S rats (*p* = 0.078) (Figure 3c). SLC27A1, also named fatty acid transport 1 (FATP1), is suggested to be important for fat deposition, promoting fatty acid esterification and lipid accumulation in adipocytes [16], being also required for fatty acid uptake and thermogenic activity in brown and beige adipose tissue [17]. Furthermore, the SADI-S group exhibited decreased levels of the master transcription factor of adipogenesis *Pparg*, despite differences not reaching statistical significance (*p* = 0.094) (Figure 3d). Since patches of small adipocytes looked more like BAT, we studied the gene expression levels of factors related to de novo beigeing of SCWAT. mRNAs encoding beige-fat-cell markers, such as *Cd137* and *Tmem26*, exhibited a tendency towards an increase in rats submitted to both bariatric surgeries, although differences did not reach statistical differences (both *p* < 0.10) (Figure 3e,f). The gene expression levels of Spp1, which also has the ability to promote the browning of WAT [18], were significantly upregulated in rats submitted to the SADI-S compared with sham-operated animals (*p* < 0.5) and exhibited a tendency towards an increase vs. SG rats (*p* < 0.1) (Figure 3g). SADI-S surgery was also associated with higher *Ucp1* transcript levels in SCWAT, but differences were not statistically significant (Figure 3j). Taken together, these data suggest that SADI-S improved SCWAT function, which was further confirmed by the decreased (*p* = 0.069) expression levels of *Atf7*, a transcription factor that inhibits BeAT biogenesis in SCWAT [19]. Interestingly, a positive correlation (r = 0.83; *p <* 0.001) between gene expression levels of *Pparg* and *Slc27a1* was detected.

### 2.4. Improved Beige Adipose Tissue Charateristics after SG and SADI-S

The increased awareness that energy-storing WAT can adopt brown-like characteristics by transforming into BeAT has kindled interest in how bariatric surgery may influence beige thermogenesis as well as energy metabolism. BeAT shows different anatomical and morphological appearance to SCWAT, sharing many features of BAT [20,21,22]. In this sense, WAT is normally characterized by an ivory or yellowish color, whereas the BeAT exhibits a reddish-brown color, with both brown and white adipocytes characteristics.

mRNA transcript levels of the beige adipocyte markers *Tmem26* (*p* < 0.05), *Cd137* (*p* < 0.01) and *Tbx1* (*p* < 0.05) were significantly increased in BeAT of rats submitted to bariatric surgery (*p* < 0.05, *p* < 0.01 and *p* < 0.05 vs. sham-operated rats) (Figure 4). Moreover, these markers were expressed at significantly higher levels after SADI-S compared SG. Despite not showing differences in *Bmp7* transcript levels, and only significant differences (*p* < 0.05) in *Prdm16* mRNA levels in SG rats, the gene expression levels of *Pgc1a*, key in the regulation of genes involved in energy metabolism, were significantly higher in BeAT of the SG and SADI-S groups (*p* < 0.05 and *p* < 0.01, respectively). A significant increase (*p* < 0.05) in mRNA expression of *Ask1*, a regulator of beige adipocyte function, was also observed in rats that underwent both bariatric surgical procedures. Finally, no differences in *Slc27a1* gene expression levels between any of the groups studied were observed, although a tendency towards increased gene expression levels of *Dio2*, essential for adaptive thermogenesis, was shown in rats submitted to both bariatric procedures. A positive correlation between mRNA levels of *Cd137* (r = 0.75; *p* < 0.01), *Tmem26* (r = 0.57; *p* < 0.05), *Pgc1a* (r = 0.58; *p* < 0.05) and *Tbx1* (r = 0.64; *p* < 0.05) with relative rectal temperature was found.

## 3. Discussion

Despite the growing understanding of energy homeostasis regulation, currently available treatments against obesity and its associated comorbidities are of generally limited long-term effectiveness, with the exception of bariatric surgery. RYGB and SG are established effective therapeutic alternatives for obesity and T2D, inducing a sustainable and marked weight loss in combination with many other health benefits [23,24,25]. Limited understanding of the mechanisms by which bariatric surgery induces and sustains weight loss is available. To our knowledge, no publications delve into the effects of SADI-S surgery on energy expenditure. Contrary to earlier reports focused on the RYGB [26,27,28], no differences in energy expenditure of SADI-S rats compared with sham-operated animals were detected. However, sleeve-gastrectomized rats exhibited increased rectal temperature compared to sham-operated counterparts, agreeing with previous studies [11]. The increased energy expenditure in animal models after gastric bypass is in agreement with some, but not all, reports in humans, and further studies are warranted in order to disentangle impact of the diverse bariatric procedures on energy expenditure in animal models and patients [27,29]. In our study, advantages of SADI-S over SG have emerged, with rats submitted to SADI-S surgery exhibiting an improved body composition as well as better control of lipid and glucose metabolism. Importantly, SADI-S induced an increase in the Adpn/lep ratio, suggesting an improved adipose tissue function after this bariatric surgical procedure.

In order to get deeper insight into the mechanistic basis of energy homeostasis changes following bariatric surgeries, we focused on the transcriptional control of the BAT and BeAT metabolic pathways. Both BAT and BeAT constitute major regulators of whole-body energy metabolism in humans and in rodents due to their ability to convert the chemical energy into heat [30]. Although no differences were observed in the positive transcriptional regulator of the brown fat cell gene program PRDM16 [31], SADI-S enhanced gene expression levels of *Bmp7*, that reportedly activates a full program of brown adipogenesis [32]. An upregulation of the essential markers of BAT function and responsible for the acquisition of the specific brown adipocyte phenotype *Pgc1a* and *Pparg* [33,34] was also identified in BAT after SADI-S. Furthermore, the histone deacetylase SIRT1, which positively acts on the activation of metabolic genes through a direct deacetylation of Pgc-1α [35], was overexpressed in BAT after SADI-S. In this regard, gene expression levels of *Sirt3*, necessary for the full acquirement of the BAT phenotype [36], were detected, together with the upregulation of *Sirt6*, which enhances AT browning and thermogenesis [37]. In line with prior findings of circulating changes in FGFs in humans and rodents [38,39], these results were concomitant with a significant increase in the expression of *Fgf21* in BAT of SADI-S rats, which also exerts strong autocrine/endocrine effects on BAT activation [40]. The improved brown adipocyte function in rats submitted to SADI-S may be also related with the increased levels of *Ask1*, although its functions and physiological relevance in AT remain controversial [18]. These data reinforce the notion of the plausible beneficial effect of the SADI-S against obesity due to increased WAT browning, rather than increased BAT activity, as evidenced by the reduced *Ucp1* and *Ucp3* gene expression levels as well as by the increased UCP1 protein levels in SCWAT. These data, together with the slight increase of the brown-adipocyte markers *Cd137* and *Tmem26* and the increased *Spp1* gene expression levels also involved in brown adipogenesis [21], suggest a stimulation of SCWAT browning. In line with this observation, an increased use of the energy stored as TG within the adipocytes, reflected by a reduction in adipocyte size, was observed together with decreased levels of *Pparg*, which also mediates HFD-induced adipocyte hypertrophy and lipid deposition [41]. The decreased levels of *Pparg* were accompanied by a congruent reduction in *Slc27a1* levels, suggesting a lower lipid storage capacity of SADI-S rats. Contrary to SADI-S rats, the activation of the endogenous classical BAT rather than the thermogenic program in the SCWAT was observed in rats submitted to SG.

Since BAT and BeAT share many morphological and biochemical characteristics, we focused on the transcript modifications in the latter tissue, observing an important upregulation of *Pgc1a* and *Ask1* gene expression levels. Moreover, increased mRNA levels of the genetic marker *Tbx1* was also found. These findings suggest, according to Abegg et al. [42], that SADI-S prevents, through WAT browning and BeAT activation, the marked temperature decrease that typically occurs during weight loss, providing a comprehensive insight into the transcriptomic modifications induced by bariatric surgery. Unlike SADI-S, SG does not involve gastrointestinal reconfiguration, so it can reasonably be assumed that it has a different effect on thermogenesis due to the different impact on the gut microbiota [43,44] linked to positively modulating BAT/BeAT thermogenesis [45,46]. Moreover, additional adipostatic factors cannot be ruled out [47,48,49].

Many studies have linked obesity to reduced energy expenditure and high RQ, due to a decreased fat oxidation [50]. The observed changing patterns of oxidative substrate in SADI-S rats, including decreased carbohydrate oxidation and increased fat oxidation, have already been described after bariatric surgery [51,52]. Furthermore, these data were corroborated by the RQ, which was decreased in rats submitted to SADI-S, with a higher weight loss, according to previous studies [53], and slightly increased in SG rats. The main reason for the decreased RQ values post-operatively is an increase in fat use and a decrease in carbohydrate consumption during *ad libitum* feeding, corroborated by GOX and LOX, and confirming that the SADI-S has shown a better efficacy regarding weight loss when compared with SG.

In addition, the efficacy of bariatric surgery in T2D improvement or remission is well documented [54]. In line with this, we observed an improvement in glucose homeostasis in SADI-S animals after 6 weeks of the surgical procedure. Moreover, obesity and co-existing metabolic conditions/comorbidities are also associated with lipid-lipoprotein abnormalities [55]. A systematic review described that randomized control trials display 76% remission of dyslipidemia after surgery [56]. In this regard, this study also attempts to compare SG and SADI-S in terms of improvement of lipid profile in DIO rats. We detected a higher reduction in plasma lipid levels after SADI-S compared to SG. SADI-S ameliorates lipid homeostasis through a decrease in serum FFA, TG and total cholesterol levels.

## 4. Conclusions

The present study indicates a fundamental difference in the amelioration of metabolic parameters between SADI-S and SG, suggesting that the manipulation of different parts of the gastrointestinal tract may lead to different physiologic effects. Our data provide new insights into the improvement of AT function after SADI-S, reinforcing the notion of the plausible beneficial effect of the SADI-S against obesity due to increased WAT browning, constituting an important mechanism that might explain the metabolic differences between surgical techniques

Additional studies assessing the impact of SADI-S on adipose tissue function in humans will be helpful to better understand the molecular mechanisms underlying the beneficial metabolic effects of this bariatric surgical procedure.

## 5. Materials and Methods

### 5.1. Experimental Animals

Four-week-old male Wistar rats (n = 56) were housed individually and maintained under controlled temperature (22 ± 2 °C), relative humidity (50 ± 10%) and on a 12:12 light-dark cycle (lights on at 08:00 am) under pathogen-free conditions. All animals had access to tap water and were maintained *ad libitum* on a normal diet (ND) (n = 10) (12.1 kJ/g: 4% fat, 82% carbohydrate and 14% protein, diet 2014S, 5 Harlan, Teklad Global Diets, Harlan Laboratories Inc., Barcelona, Spain) or a high-fat diet (HFD) (23.0 kJ/g: 60% fat, 27% carbohydrate and 14% protein; diet F3282, Bio-Serv, Frenchtown, NJ, USA) for 4 months before being randomly selected to undergo a SADI-S (n = 12), a SG (n = 11) or sham surgery (n = 13). The HFD is associated with the development of an obesogenic metabolic state. Following the surgical intervention, the animals were fed a ND except 10 rats, which were fed ad libitum the HFD in order to maintain the obese phenotype. Body weight and food intake were recorded weekly to monitor the progression of obesity. All procedures for animal use conformed to the European Guidelines for the Care and Use of Laboratory Animals (directive 2010/63/EU), and the study was approved by the Ethical Committee for Animal Experimentation of the University of Navarra (026/19).

### 5.2. Surgical Procedures

After an overnight fasting period, obese rats were anesthetized with isoflurane (Laboratorios Esteve, Barcelona, Spain; 4% for induction and 2–2.5% for maintenance) administered in a mixture with oxygen at a constant flow of 0.5–0.7 L/min through a cone mask for rodents. All surgeries were performed under sterile conditions, and the rat abdominal wall was shaved and swabbed with povidone-iodine solution. Antibiotic prophylaxis with intramuscular administration of 25 mg/kg body weight of enrofloxacine (Laboratorios Esteve) was applied. Thirty-six rats underwent surgical protocols consisting of either SADI-S (n = 12), SG (n = 11) or a sham operation (n = 13) as detailed below.

SG was conducted according to previously described procedures [9,57]. Briefly, an upper median 3.5–4 cm middle incision was performed; the gastrosplenic ligament was divided, and the stomach was externalized by placing it on a saline-moistened gauze pad. The AutoSuture TA automatic stapler (DCT Series, Tyco healthcare Group LP, Norwalk, CT, USA) with a TA30V3L (three rows; 2.5 mm) load was carefully placed along the greater curvature from the antrum to the fundus, leaving the pylorus intact and a tubular gastric remnant in continuity, maintaining the free passage of the food from the esophagus to the duodenum and drastically reducing the stomach volume by about 60–70% (Figure 5a). The SG-sham procedure involved analogous stomach isolation, manipulation and replacement in the same position in the abdomen. SG had a mortality rate of 0% during the first two days post-surgery. SADI-S surgery was performed according to previously described methodology [16]. After a short midline abdominal incision of 3.5–4 cm length, the gastrosplenic ligament was divided, the connective tissues surrounding the stomach as well as the gastrohepatic and the gastrosplenic ligaments were dissected using cotton swabs, and the stomach was externalized by placing it on a saline-moistened hot gauze pad. A stomach pouch of reduced volume was created as described for sleeve gastrectomy [12,15]. Briefly, the AutoSuture TA automatic stapler load was carefully placed along the greater curvature from the antrum to the fundus, leaving the pylorus intact and a tubular gastric remnant in continuity, maintaining the free passage of the food from the esophagus to the duodenum and drastically reducing the stomach volume by about 60–70%. The lesser curvature was dissected; the great curvature along with the gastric fundus and corpus was removed, and the vascular supply was isolated in this region. Subsequently a 10 mm clip was placed in the distal duodenum, being cut proximally and verifying the vascularization of its edges. The small intestine was externalized on a saline-moistened gauze pad, and the cecum and terminal ileum were identified in order to determine, 35 cm distally from the ileocecal valve, the point of the distal ileum to be anastomosed with the duodenum. After creating an incision of the same length as the proximal duodenum diameter, the ileum was ascended and anastomosed to the gastric pouch 0.5 cm from the pylorus using an intermittent stitch of 6–0 Prolene suture. Before closure, the tubular gastric remnant was fixed to the abdominal wall with a suture to avoid any twisting. This procedure permits the bypass of the entire duodenum and distal ileum, transiting that ingested food from the esophagus to the gastric tube and then directly to the distal ileum (Figure 5b). This method deviates from the classical One-Anastomosis Gastric Bypass procedure, which traditionally has a longer limb length than that of RYGB surgery. Sham-operated rats underwent similar incisions, operative conditions and time, without intestinal excisions.

After SADI-S and SG surgery, hemostasis was verified; peritoneal cavity was cleaned with saline solution, and the abdominal wall was closed with running 3/0 polyglycolic acid suture including the peritoneum and aponeurotic muscle layers. All animals were administered subcutaneously 5 mL saline to avoid dehydration as well as 0.03 mg/kg buprenorphine (Schering-Plough, Madrid, Spain) as analgesic measure. Rats consumed a liquid diet with 5% glucose and 0.9% saline solution liquid diet for the first 3 post-operative days and were transitioned back to solid diet by day 5. The survival rate of SG and SADI-S rats was 91% and 75%, respectively.

Six weeks after the surgical interventions, overnight fasting rats were sacrificed by decapitation and serum and plasma samples were obtained by cold centrifugation (4 °C) at 700× *g* for 15 min and stored at −80 °C. Epididymal (EWAT), subcutaneous (SCWAT), perirenal (PRWAT) white AT and BAT depots were carefully excised. BeAT within the inguinal SCWAT was detected by its reddish–brown color and, after its identification, the depot was carefully excised, removing all remnants of SCWAT. Tissue samples were immediately frozen in liquid nitrogen and stored at −80 °C.

### 5.3. Temperature Measurements

Body temperature was determined by thermometry using a rectal thermoprobe coupled to a digital thermometer (YSI 4600 Series Precision Thermometers, YSI Temperature, Dayton, OH, USA). Furthermore, the surrounding interscapular BAT skin temperature was measured using an infrared camera (E60bx Compact-Infrared-Thermal-Imaging-Camera; FLIR Systems, West Malling, Kent, UK) and analyzed with the specific associated software (FLIR-Tools-Software; FLIR Systems, Inc.) Mean skin temperature surrounding interscapular BAT was calculated as the average temperature in defined interscapular areas (2 cm Ø) recorded by analyzing each picture.

### 5.4. Oxygen Consumption and Daily Energy Expenditure

Different metabolic parameters, including VO_2_, VCO_2_ and RQ were assessed using the Oxylet Physiocage System (Panlab, Barcelona, Spain) and the software suite METABOLISM (V2.2.01, Panlab). The Oxylet system consists in a commercially available system that study metabolism in rodents through an open-circuit indirect calorimetry. The RQ was calculated as VCO_2_/VO_2_; RQ~1 corresponds to carbohydrate oxidation, whereas RQ ~ 0.7 corresponds to fat oxidation. The daily energy expenditure (EE) was determined according to the formula EE [kcal/(day × kg 0.75)] = VO_2_ × 1.44 × [3.815 + (1.232 × RQ)]. The data for VO_2_ and VCO_2_ are expressed as consumed volume of oxygen per minute per kilogram body weight (mL/min/kg 0.75). GOX and LOX were calculated according to the following equations: GOX = (4.57 × VCO_2_ − 3.23 × VO_2_) × (3.74 × 4.186/60) (W) and LOX = (1.69 × VO_2_ − 1.69 × VCO_2_) × (9.46 × 4.186/60) (W) [58].

### 5.5. Blood Measurements

Serum concentrations of TG, total cholesterol (Infinity, Thermo Electron Corporation, Melbourne, Australia), FFA (Wako Chemicals, GmbH, Neuss, Germany) and glycerol (Sigma, St. Louis, MO, USA) were quantified by enzymatic methods using commercially available kits in blood samples obtained under fasting conditions. Blood glucose was measured using an automatic glucose sensor (Ascensia Elite, Bayer, Barcelona, Spain). Leptin, insulin and adiponectin were assessed by ELISA (Crystal Chem, Inc., Chicago, IL, USA) as previously described [59,60,61] Intra- and inter-assay coefficients of variation for measurements were 4.5% and 5.4%, 3.5% and 6.3% and 2.6% and 5.3%, respectively. To estimate insulin resistance, the HOMA index as fasting insulin concentration (μU/mL) × fasting glucose concentration (mmol/L)/22.5 was calculated. An indirect measure of insulin sensitivity was determined by using the quantitative insulin sensitivity check index (QUICKI) as follows: 1/[log(fasting insulin μU/mL) + log(fasting glucose mg/dL)]. The adipocyte insulin resistance (Adipo-IR) index, as a surrogate of adipocyte dysfunction, was calculated as fasting FFA (mmol/L) × fasting insulin (pmol/L) [62].

### 5.6. RNA Extraction and Real-Time PCR

Total RNA was extracted from SCWAT, BAT and BeAT samples by homogenization with an ULTRA-TURRAX^®^ T 25 basic (IKA^®^ Werke GmbH, Staufen, Germany) using QIAzol^®^ Reagent (Invitrogen, Barcelona, Spain). Samples were purified with the RNeasy Lipid Tissue Mini kit (Qiagen) and treated with DNase I (RNase-free DNase Set, Qiagen). The RNA integrity and quantification were performed with the NanoPhotometer (Implen GmbH, München, Germany). For first strand cDNA synthesis constant amounts of 2 μg of total RNA were reverse transcribed in a 40 μL final volume using random hexamers (Roche Molecular Biochemicals, Mannheim, Germany) as primers and 400 units of M-MLV reverse transcriptase (Invitrogen, Carlsbad, CA, USA) as described earlier [19]. The transcript levels were quantified by Real-Time PCR (7300 Real Time PCR System, Applied Biosystem, Foster City, CA, USA). Both primers and probes (Appendix A) were designed using the software Primer Express 2.0 (Applied Biosystems, Foster City, CA, USA). All results were normalized to the levels of *18S* rRNA (Applied Biosystems), and relative quantification was calculated using the ΔΔCt formula. Relative mRNA expression was expressed as fold expression over the calibrator sample (average of gene expression corresponding to the ND or sham group) [63].

### 5.7. Determination of Adipocyte Surface Area

The adipocyte surface area (CSA) of adipocytes in SCWAT was measured as previously described [64]. Briefly, sections (6 µm) of 4% formaldehyde-fixed and paraffin-embedded SCWAT biopsies were stained with H&E. Images of three fields per section from each animal were captured with the 40× objective, and the adipocyte CSA from, at least, 100 cells/section were measured using AxioVision Release 4.6.3 software (Zeiss, Göttingen, Germany).

### 5.8. Immunohistochemistry of UCP-1

The immunohistochemistry was carried out in SCWAT using the indirect immunoperoxidase method [65] using rabbit monoclonal anti-UCP-1 antibodies (Abcam) diluted 1:100 in Tris-buffer saline. Negative control slides without primary antibody were included for the assessment of non-specific staining.

### 5.9. Statistical Analysis

Data are expressed as the mean ± S.E.M. Unpaired Student’s *t* test and one-way ANOVA followed by Bonferroni *post hoc* tests were used for mean comparison between the groups. Pearson’s correlation coefficients (r) were used to analyze the association between variables. Gene expression levels and variables with a non-normal distribution were logarithmically transformed. Statistics were calculated by the SPSS/Windows version 15.0 software (SPSS, Inc., Chicago, IL, USA), and the figures were created with GraphPad Prism version 8.3 (GraphPad Software, Inc., San Diego, CA, USA). A *p* value less than 0.05 was considered statistically significant.

## Figures and Tables

**Figure 1 ijms-23-11641-f001:**
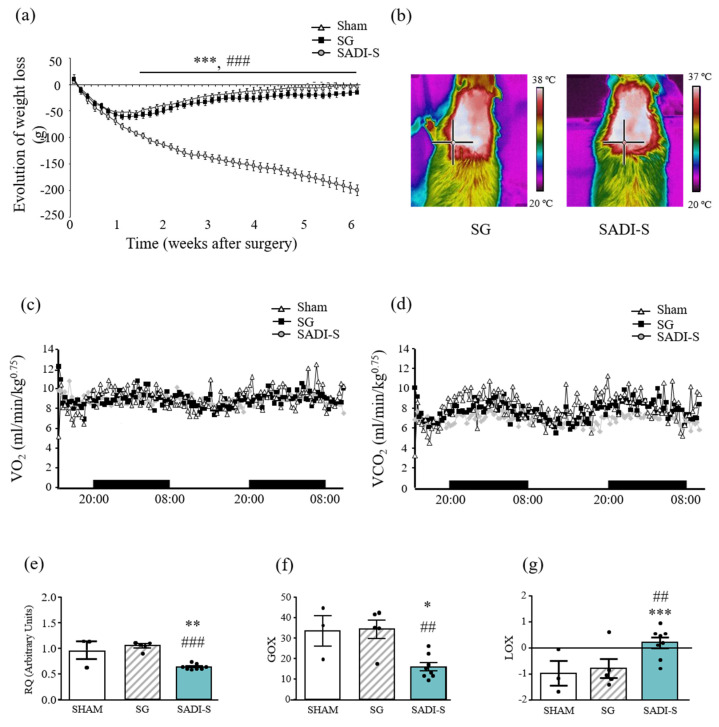
Body weight and energy expenditure after SG and SADI-S. (**a**) Growth curves show the evolution of body weight of rats 6 weeks after surgeries. (**b**) Representative thermal images of the effect of SG and SADI-S on interscapular BAT temperature are also illustrated. Curves show the evolution of (**c**) oxygen consumption and (**d**) carbon dioxide production analyzed by indirect calorimetry. (**e**) Respiratory quotient (RQ), (**f**) glucose oxidation (GOX) and (**g**) lipid oxidation (LOX) values are shown. Values are the mean ± SEM (n = 9–13 per group). Differences between groups were analyzed by one-way ANOVA followed by Bonferroni *post hoc* tests. * *p* < 0.05, ** *p* < 0.01; *** *p* < 0.001 vs. sham-operated group; ## *p* < 0.01, ### *p* < 0.001 vs. SG group.

**Figure 2 ijms-23-11641-f002:**
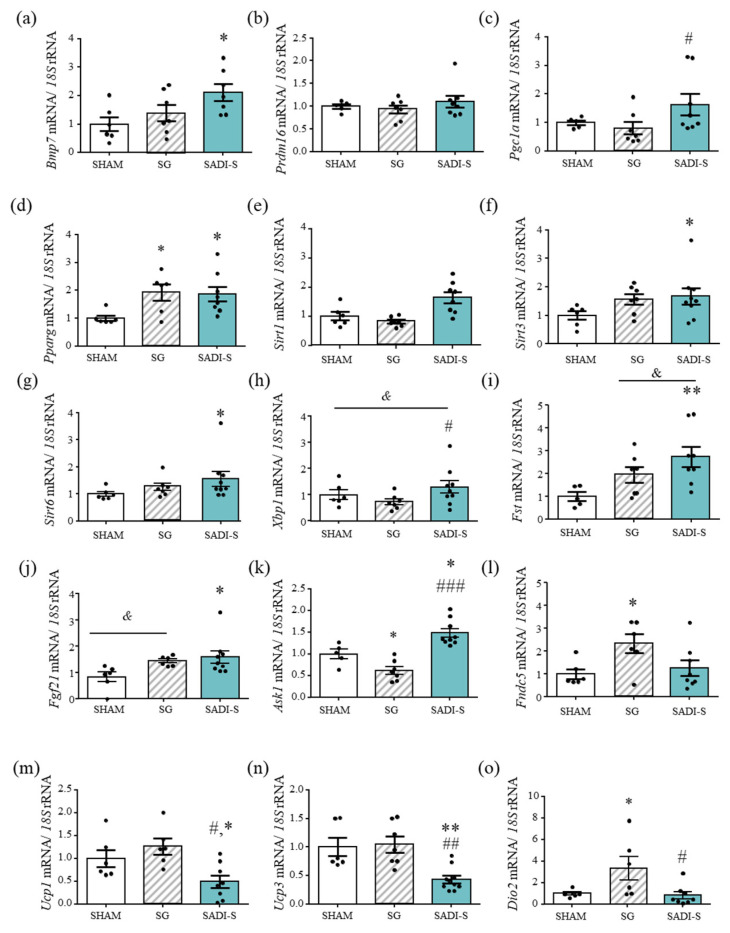
Expression of genes involved in BAT differentiation, function and regulation in DIO rats six weeks after bariatric surgery. (**a**–**o**) Bar graphs show the expression levels of genes related to BAT differentiation and function. mRNA data were normalized for the expression of *18S* rRNA. The expression in sham-operated rats was assumed to be 1. Values are the mean ± SEM (n = 6–8 per group). Differences between groups were analyzed by one-way ANOVA followed by Bonferroni *post hoc* tests. * *p* < 0.05, ** *p* < 0.01 vs. sham surgery; # *p* < 0.05, ## *p* < 0.01, ### *p* < 0.001 vs. SG.

**Figure 3 ijms-23-11641-f003:**
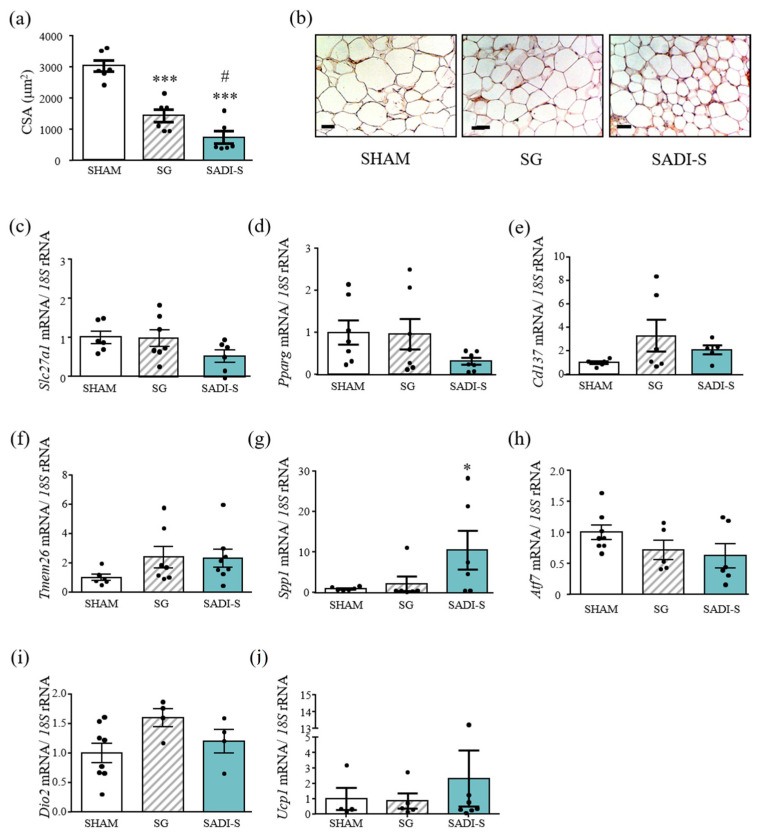
Gene and protein expression levels of genes involved in SCWAT function and browning. (**a**) Bar graphs show the cell surface area (CSA) of subcutaneous adipocytes. (**b**) Immunostaining of UCP-1 in SCWAT of the experimental animals. Magnification X200 (scale bar = 50 µm). (**c**–**j**) Expression levels of genes related to SCWAT function are also represented. mRNA data were normalized for the expression of *18S* rRNA. The expression in sham operated rats was assumed to be 1. Values are the mean ± SEM (n = 6 per group). Differences between groups were analyzed by one-way ANOVA followed by Bonferroni *post hoc* tests. *** *p* < 0.001 vs. sham-operated group; # *p* < 0.05 vs. SG rats.

**Figure 4 ijms-23-11641-f004:**
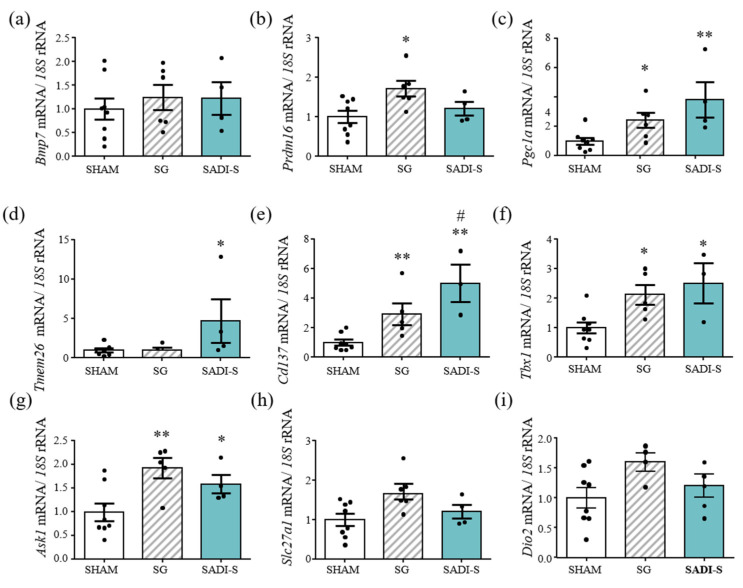
Expression of genes related to beige adipose tissue function in BeAT of rats after bariatric surgery (**a**–**i**). Bar graphs show the mRNA expression levels of factors related to beige adipose tissue function. mRNA data were normalized for the expression of *18S* rRNA. The expression in sham-operated rats was assumed to be 1. Values are the mean ± SEM (n = 6 per group). Differences between groups were analyzed one-way ANOVA followed by Bonferroni *post hoc* tests. * *p* < 0.05, ** *p* < 0.01 vs. sham-operated group; # *p* < 0.05 vs. SG group and *p* < 0.1.

**Figure 5 ijms-23-11641-f005:**
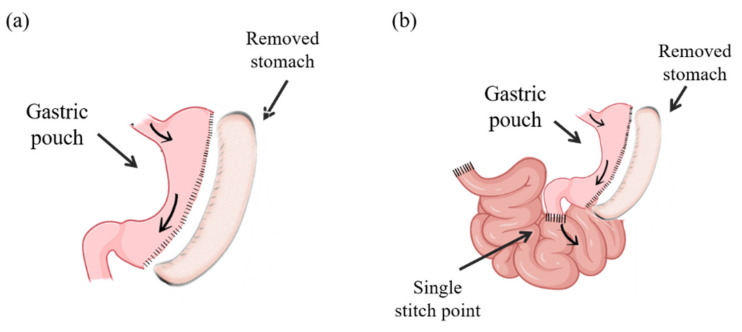
Schematic representation of (**a**) sleeve gastrectomy (SG) and (**b**) single anastomosis duodeno-ileal bypass with sleeve gastrectomy (SADI-S).

**Table 1 ijms-23-11641-t001:** Body composition, body and BAT temperature and metabolic profile 6 weeks after SG and SADI-S.

Determination	Sham Surgery (n = 13)	SG (n = 10)	SADI-S (n = 9)	*p*
Final body weight (g)	529 ± 8	510 ± 7	345 ± 18 ^c,f^	**<0.001**
Rectal temperature (°C)	36.5 ± 0.1	37.0 ± 0.1 ^b^	36.7 ± 0.1	**0.010**
BAT temperature (°C)	37.5 ± 0.2	38.1 ± 0.1 ^a^	37.6 ± 0.2	**0.044**
VO_2_ (mL/min/kg^0.75^)	8.72 ± 0.10	8.69 ± 0.06	8.68 ± 0.04	0.979
VCO_2_ (mL/min/kg^0.75^)	9.59 ± 0.12	9.09 ± 0.07 ^b^	8.34 ± 0.04 ^c,f^	**<0.001**
Epididymal WAT (g/100 g BW)	1.19 ± 0.07	1.31 ± 0.10	0.39 ± 0.16 ^c,f^	**<0.001**
Subcutaneous WAT (g/100 g BW)	0.92 ± 0.07	0.94 ± 0.08	0.29 ± 0.08 ^c,f^	**<0.001**
Perirenal WAT (g/100 g BW)	1.30 ± 0.10	1.31 ± 0.08	0.40 ± 0.18 ^c,f^	**<0.001**
BAT (g/100 g BW)	0.157 ± 0.017	0.145 ± 0.007	0.056 ± 0.008 ^c,f^	**<0.001**
Total WAT (g/100 g BW)	3.41 ± 0.21	3.44 ± 0.16	0.91 ± 0.37 ^c,f^	**<0.001**
Glucose (mg/dL)	73 ± 2	79 ± 2	59 ± 3 ^c,f^	**<0.001**
Insulin (ng/mL)	3.6 ± 0.5	2.9 ± 0.4	1.4 ± 0.2 ^b,e^	**<0.01**
HOMA	0.7 ± 0.1	0.7 ± 0.1	0.2 ± 0.1 ^b,e^	**<0.001**
QUICKI	0.43 ± 0.02	0.42 ± 0.01	0.53 ± 0.01 ^c,f^	**<0.001**
FFA (mg/dL)	19 ± 1	23 ± 1	18 ± 2 ^d^	0.084
TG (mg/dL)	118 ± 14	93 ± 8	51 ± 8 ^b,d^	**<0.001**
Cholesterol (mg/dL)	56 ± 3	51 ± 4	36 ± 4 ^b,d^	**0.003**
Glycerol (mg/dL)	0.019 ± 0.002	0.018 ± 0.001	0.015 ± 0.001	0.115
Adipo-IR index	26.2 ± 0.2	28.2 ± 0.2	8.3 ± 0.1 ^a,e^	**0.002**
Leptin (ng/mL)	4.3 ± 0.4	3.3 ± 0.4	0.4 ± 0.1 ^c,f^	**<0.001**
Adiponectin (ng/mL)	9.8 ± 0.4	11.4 ± 0.6	6.5 ± 0.8 ^c,f^	**<0.001**
Adpn/leptin ratio	2.50 ± 0.21	3.24 ± 0.34	15.98 ± 3.01 ^c,f^	**<0.001**

SG, sleeve gastrectomy; SADI-S, single anastomosis duodenoileal bypass with sleeve gastrectomy; WAT, white adipose tissue; BAT, brown adipose tissue; BW, body weight; HOMA, homeostasis model assessment; QUICKI, quantitative insulin sensitivity check index; FFA, free fatty acids; TG, triglycerides; Adipo-IR, adipose tissue insulin resistance index; Adpn, adiponectin. Values presented as the mean ± S.E.M. Differences between groups were analyzed by one-way ANOVA followed by Tukey’s post-hoc test. ^a^
*p* < 0.05, ^b^
*p* <0.01, ^c^
*p* < 0.001 vs. sham-operated group; ^d^
*p* < 0.05, ^e^
*p* < 0.01, ^f^
*p* < 0.001 vs. SG group. Bold lettering indicates statistically significant values.

## Data Availability

The data presented in this study are available upon reasonable request from the corresponding author. The data are not publicly available due to privacy restrictions.

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
