# Peer review of "Improved Adipose Tissue Function after Single Anastomosis Duodeno-Ileal Bypass with Sleeve-Gastrectomy (SADI-S) in Diet-Induced Obesity"

_ijms, 2022, doi:10.3390/ijms231911641_

Round 1
Reviewer 1 Report
In the manuscript entitled “Improved adipose tissue function after Single Anastomosis Duodeno-ileal Bypass with Sleeve-Gastrectomy (SADI-S) in diet-induced obesity”, the author compared the effects of Sleeve Gastrectomy (SG) and SADI-S on the metabolic profile of diet-induced obese rats and concluded that SADI-S surgery was associated with decreased fat mass, increased energy expenditure, and improved lipid and glucose metabolism when compared to the SG counterparts. The finding is interesting, but the paper suffers from a large number of problems, which detracts from the overall quality. The major issue is that the data presented in the manuscript can’t support and even contradict the conclusion.
- The description in lines 91 to 93 “Neither liver nor skeletal muscle weights were statistically different among groups suggesting that the decreased body weight was not due to fat-free mass loss” is inappropriate. It would be better to measure the body composition (total fat mass and lean mass) to get such a conclusion.
- The result in lines 103 to 104 “Energy expenditures of SADI-S rats showed a significant increase compared to Sham and SG counterparts….” is not shown in Table1.
- The representative thermal images in Fig1b can’t reflect the change in BAT temperature after SG and SADI-S surgery, it would be better to add the quantification.
- Fig1c and 1d indicated the decreased energy expenditure in the SADI-S group, which is inconsistent with the conclusion that SADI-S surgery was associated with increased energy expenditure. Please justify it.
- Please provide the quantification and statistical analysis for Fig1c and 1d.
- The description in lines 141 to 143 “Transcript levels of Bmp7, a factor involved in the activation of the brown adipogenesis program was significantly increased after SADI-S” is inappropriate, as there is no significant difference in BMP7 expression between SG and SADI-S groups.
- The decreased expression of UCP1 and DIO2 in the SADI-S group indicated the decreased thermogenic activity of brown adipose tissue, which is inconsistent with the conclusion of the manuscript. Please provide an explanation.
- Please provide the figures for the positive correlation between mRNA levels and rectal temperature in lines 164 to 166.
- The immunohistochemistry staining of UCP1 in Fig3b is unconvincing. Beige adipocyte marker genes are also comparable between SG and SADI-S groups in Fig3e, 3f, and 3i, which is inconsistent with the morphological change of iWAT, please provide an explanation.
- The description of the Slc27a1 in lines 182 to 185 is inappropriate, as Slc27a1 is defined as a beige-selective signature gene in iWAT, which is required for fatty acid uptake and thermogenic activity in brown and beige adipose tissue.
- For Fig 4, please provide more information about the BeAT, such as how to distinguish the scWAT and BeAT in SG and SADI-S rat models.
- It would be better to include more information about the blood measurements in line 410, is the serum collected under fasting conditions?
Reviewer 2 Report
In this work, the authors studied the metabolic effects and their impact on white (sWAT), brown (BAT) and beige (BeAT) subcutaneous adipose tissue function of Sleeve Gastrectomy (SG) and single anastomosis Duodenoyl Bypass with SG (SADI-S) in an animal model of diet-induced obesity (DIO).
The results obtained suggest that SADI-S surgery shows superior efficacy compared with SG in weight loss through improved energy expenditure and that increased BAT/BeAT activation may be an important mechanism explaining the metabolic differences between the two bariatric surgical techniques.
The article is interesting, well-organized and well-written, and adds new evidences on the investigated field.
Points
I believe that a figure representing the two different surgical technique could be of interest for the readers, considering that the IJMS is a journal that targets a broad audience of researchers.
Round 2
Reviewer 1 Report
The authors have addressed some of my concerns in the revised manuscript. However, the major issue is that the data presented in the manuscript can’t support and even contradict the conclusion, especially the decreased energy expenditures, decreased body temperature, and decreased UCP1 expression in BAT of SADI-S rats are inconsistent with the major conclusion described in the manuscript “Our findings provide evidence that the enhanced metabolic improvement and their continued durability after SADI-S compared to SG relies, at least in part, on the improvement of BAT and BeAT phenotype and function”.
Still, I have other comments shown below that need to be clarified.
- In Table 1, it’s inappropriate to use the relative rectal temperature and relative BAT temperature (normalized to the body weight) to indicate the activation of BAT.
- The decreased respiratory quotient (RQ) in Figure 1e cannot reflect the improved energy expenditure after SADI surgery. Actually, the decreased VO2 in Table 1 and Figure 1d, decreased BAT temperature in Figure 1b and decreased UCP1 in Figure 2m indicated the decreased energy expenditure after SADI surgery.
- The immunohistochemistry staining of UCP1 in Fig3b is unconvincing, especially the positive signaling in the sham group.
- The beige adipose tissue (BeAT) is an inducible form of adipose tissue that is located in subcutaneous inguinal white adipose tissue. Does the author use the same fat depot for the gene expression analysis shown in figure 3 (scWAT)and figure 4 (BeAT)?
Round 3
Reviewer 1 Report
The authors have answered my questions and the paper has been greatly improved.
Author Response
Dear Reviewer,
We would like to express our gratitude. Your comments are have served to substantially improve the manuscript.
Thank you in advance for your consideration.
Yours sincerely,
Sara Becerril, PhD
Metabolic Research Laboratory
Clínica Universidad de Navarra
Avda Pío XII 36
31008 Pamplona, Spain
Tel: +34 948 25 54 00 (ext. 5133)
Fax: +34 948 29 65 00
e-mail: sbecman@unav.es